# Comparing the impact of genotypic based diagnostic algorithm on time to treatment initiation and treatment outcomes among drug-resistant tuberculosis patients in Amhara region, Ethiopia

**Getahun Molla Kassa** , **Mehari Woldemariam Merid** *, **Atalay Goshu Muluneh, Haileab Fekadu Wolde**

Department of Epidemiology and Biostatistics, Institute of Public Health, College of Medicine and Health Sciences, University of Gondar, Gondar, Ethiopia

* mehariho19@gmail.com

**Data Availability Statement:** The data cannot be shared publicly as contains some seemingly

## Abstract

### Background

To end Tuberculosis (TB) by 2030, early detection and timely treatment of Drug-Resistant Tuberculosis (DR-TB) is vital. The role of rapid, accurate, and sensitive DR-TB diagnostic tool is indispensable to accelerate the TB control program. There are evidence breaks in the time difference and its effect on treatment outcomes among different DR-TB diagnostic tools in Ethiopia. This article aimed to compare the different DR-TB diagnostic tools with time pointers and evaluate their effect on the treatment outcomes.

### Method

We performed a retrospective chart review of 574 DR-TB patients from September 2010 to December 2017 to compare the impact of molecular DR-TB diagnostic tests (Xpert MTB/RIF, Line Probe Assay (LPA), and solid culture-based Drug Susceptibility Testing (DST)) on time to diagnosis, treatment initiation, and treatment Outcomes. Kruskual-Wallis test was employed to assess the presence of a significant difference in median time among the DR-TB diagnostic tests. Chi-Square and Fisher exact tests were used to test the presence of relations between treatment outcome and diagnostic tests.

### Result

The data of 574 DR-TB patients were included in the analysis. From these, 321, 173, and 80 patients were diagnosed using Xpert MTB/RIF, Line Probe Assay (LPA), and solid culture-based DST, respectively. The median time in a day with (Interquartile range (IQR)) for Xpert MTB/RIF, LPA, and solid culture-based DST was from a first care-seeking visit to diagnosis: 2(0, 9), 4(1, 55), and 70(18, 182), from diagnosis to treatment initiation: 3(1, 8), 33(4, 76), and 44(9, 145), and from a first care-seeking visit to treatment initiation: 4(1, 11), 3(1, 12)

unidentifier alone may have potentially identifying information in combination that will have ethical consequences. We have two points of contact for data requests: 1. Any interested body can communicate the lead author for data access after which the lead author will communicate to the IRB of University of Gondar and provide the data. 2. One can access data by requesting the non-author contact, the academic coordinator and member of the IRB of the college of medicine and health sciences at the University of Gondar below with the contact address of: demissma1623@gmail.com.

**Funding:** The authors received no specific funding for this work.

**Competing interests:** The authors have declared that no competing interests exist.

and 76(3.75, 191) respectively. The shorter median time was observed in the Xpert MTB/RIF followed by the LPA, and this was statistically significant with a p-value <0.001. There was no statistically significant difference concerning treatment outcomes among the three DST tests.

## Conclusion

Xpert MTB/RIF can mitigate the transmission of DR-TB significantly via quick diagnosis and treatment initiation followed by LPA as equating to the solid culture base DST, particularly in smear-positive patients. However, we didn't see a statistically significant impact in terms of treatment outcomes. Xpert MTB/RIF can be used as the first test to diagnose DR-TB by further complimenting solid culture base DST to grasp the drug-resistance profile.

## Introduction

Globally, Tuberculosis (TB) is one of the leading causes of mortality in 2018 [1]. Drug-Resistant Tuberculosis (DR-TB) continues to be one of the challenging global public health crises with more than half a million new cases annually [1, 2]. The World Health Organization (WHO) reported that the global prevalence of DR-TB was 18% among previously treated and 3.4% among newly treated TB cases while in Ethiopia the estimated DR-TB was 0.71% of new cases and 16% of previously treated cases [1]. The crisis of DR-TB progressively jeopardizes the TB control program in Ethiopia. Based on the 2017/2018 Ethiopian national TB reference laboratory data, the proportion of Multidrug-Resistant Tuberculosis (MDR-TB) was 11.6% [3].

Improving the detection time and diagnostics modalities was a game-changer in the control of the MDR-TB burden [1, 4]. A study in Russia pointed out that Line Probe Assay (LPA) compared to solid (Löwenstien-Jensen or LJ) culture, significantly shorten time to diagnosis, and reduced delay to treatment among DR-TB, and improve final treatment outcomes [5]. Findings from Cape Town South Africa point out that Xpert MTB/RIF assay has shortened the time of diagnosis and treatment initiation as compared to LPA [6]. But as evidenced from studies conducted in high burden countries, patients who were diagnosed by Xpert MTB/RIF had better treatment success and shorter time of anti-TB treatment initiation compared to patients diagnosed by culture or LPA [7, 8]. Xpert MTB/RIF assay is highly sensitive, specific, and comparable to the gold standard (solid culture) Drug Sensitivity Tests (DST) method for the diagnosis of DR-TB with a shorter time of detection and low biohazard [9]. Xpert MTB/RIF assay is a rapid, efficient, and reliable technique for the diagnosis of DR-TB with high sensitivity and specificity to detect clinically suspected DR-TB cases [10]. Other scholars in Ethiopia have also documented that Xpert MTB/RIF could be used to detect DR-TB cases effectively with shorter time of initiation of anti-TB treatments among smear or solid culture-positive cases [11]. A study conducted in Iran recommends Xpert MTB-RIF could be used as an early diagnostic method whose results must be confirmed by the standard proportional method as being quick and helpful [12]. As compared to solid culture, Xpert MTB/RIF had 99.1% sensitivity and 96% specificity to detect TB cases [13]. Molecular techniques of DR-TB diagnosis could shorten treatment initiation time and reduce the risk of unfavorable outcome [7]. Studies in South Korea point that Xpert MTB/RIF was an accurate technique to diagnose rifampicin-resistant TB (RR-TB) and reduces the risk of transmission by shortening the time of

culture conversion [14]. Findings from South Africa indicated that Xpert MTB/RIF assay improves time to treatment initiation but doesn't improve MDR-TB treatment outcomes compared to patients diagnosed with smear/culture cohorts [6, 15]. Compared to conventional culture and DST tests, Xpert MTB/RIF shortens result turn-around-time and excellent concordance to detect RR-TB [16, 17] and early treatment initiation [6, 8]. It has been reported that Xpert MTB/RIF enhances DR-TB case detection among pediatrics age groups in India [18] and could detect a large number of people with TB that routine services failed to detect [19]. Different studies showed that using Xpert MTB-RIF assay for the diagnosis of DR-TB could increase case detection rate in diagnostic centers [9–11, 13, 20–25]. And controversies were seen on the case of treatment outcome when using Xpert MTB/RIF.

World Health Originations, proposed rapid detection, and early initiation of therapy are a key strategy for DR-TB control and treatment outcomes. To the best of our knowledge, no study has been conducted in the study area before. We aim to assess the impact of a Genotypic Based diagnostic algorithm on time to treatment initiation and treatment outcomes for DR-TB patients in the Amhara region, Northwest Ethiopia. This might give important clues for the researchers and policymakers of the region.

## Methods

### Study design and period

This was an institutional-based retrospective comparative observational study, conducted from September 2010 to December 2017.

### Settings and populations

The source population was all DR-TB patients in the Amhara region while the study population were DR-TB patients registered and follow their treatment in the four (University of Gondar, Boru-Meda, Debre-Markos, and Woldia) DR-TB treatment centers of Amhara Regional State, Ethiopia. Over 90% of DR-TB patients in the region started and follow their Second Line ant-TB Drugs (SLD) in these four hospitals [26]. University of Gondar Comprehensive Specialized Hospital is the second-largest hospital giving clinical care and management for DR-TB in the Country, which is found in the Central Gondar zone of Amhara. Boru-Meda generalized hospital is one of the oldest hospitals known to give special care for TB and Leprosy patients historically, now it advanced its care for DR-TB patients, found in the South Wello zone of the Amhara. Debre-Markos referral hospital is serving DR-TB patients in the Easter Gojjam Zone of Amhara that serves patients coming from the catchment. The fourth hospital is Woldia general hospital which is found in the North Wello Zone of Amhara and serves patients from the Northeaster parts of Ethiopia. All four hospitals give services not only for patients who come from their catchment area but also to those coming from the neighboring Regional states (Tigray, Afar, and Benshagul Gumuz) of Ethiopia. Almost all of the DR-TB patients had enrolled in SLD treatment with a bacteriologically confirmed result of a list resistant to rifampicin (RR, MDR, Pre-XDR, or XDR) TB (Fig 1). Culture-based DST and LPA were used alternatively to diagnose DR-TB in Ethiopia to the time 2013. Later in 2013, Xpert MTB/RIF was introduced in the study area to diagnose RR-TB. Currently, only two TB culture-based DST and LPA sites are available in the region, which is located in Bahr-Dar and Gondar. The access of these three diagnosis tests varies from site to site, because of this and some other reasons even after the introduction of Xpert MTB/RIF, culture and LPA were also used as the first test to diagnose DR-TB. Currently, those three tests are done serially. Once the patient was diagnosed to have RR-TB by Xpert MTB/RIF then first and the second line LPA was performed to

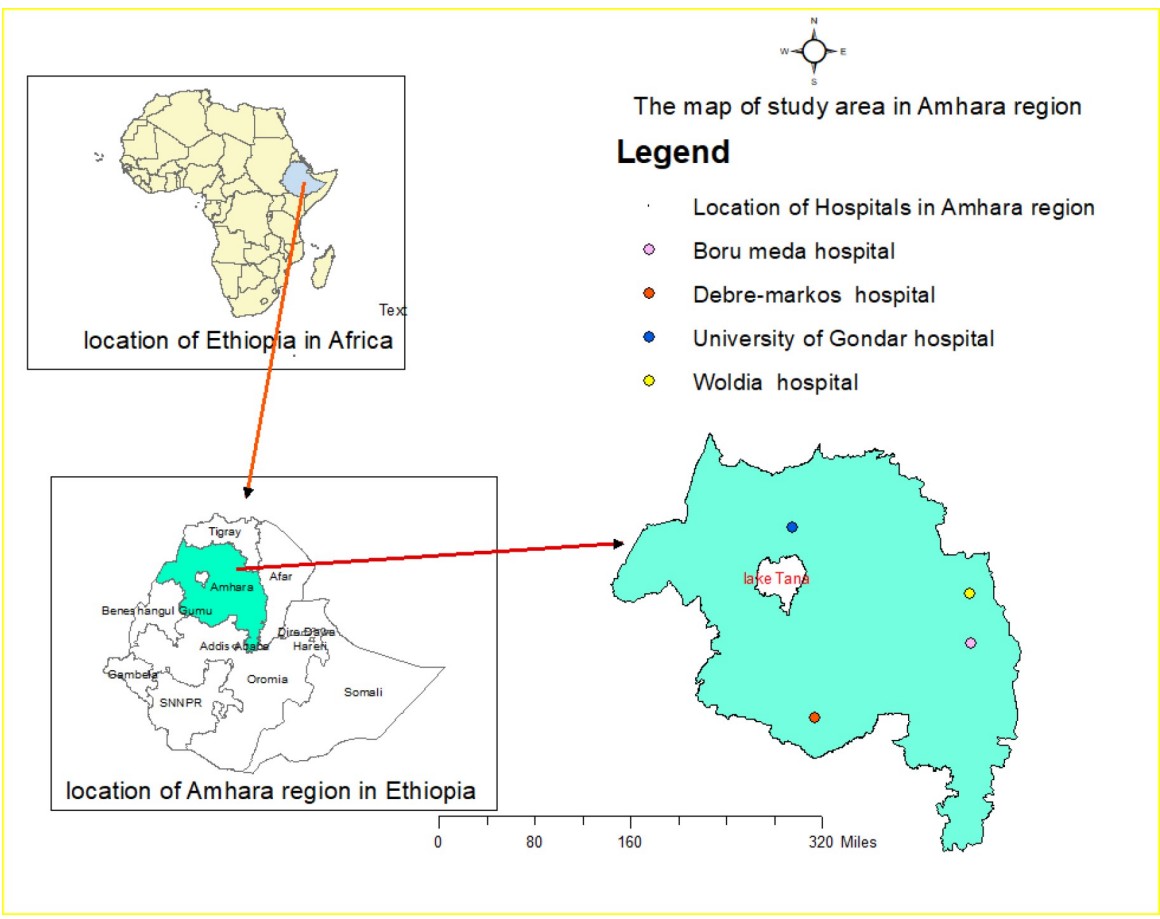

**Fig 1. Map of the study area indicating the locations of the four hospitals.**

see further resistance for Isoniazid, Floroqunolols, and injectable ant-TB drugs, and culture was done to monitor the treatment response and when further phenotypic DST is necessary.

## Variable of the study

Time from a first care-seeking visit to diagnosis was defined as the total date taken from the patient visit the health facility for care to the time of their TB diagnosis; time from diagnosis to treatment initiation was the date difference from the DR-TB diagnosis to initiation of anti-TB treatment; and time from first care-seeking visit to treatment initiation was labeled as the date from the patient comes to the health facility for seeking health care to the date the patient start anti-TB treatment. **Treatment outcome** of patients was dichotomized into two and defined as successful when the patient becomes a cure and completed the recommended treatments without showing signs and symptoms of TB. While, those patients who have died, lost to follow-up, and have a failure treatment outcome were recorded as an unsuccessful treatment outcome [27].

## Sample size and sampling procedures

All DR-TB patients registered from September 2010 (the start of DR-TB treatment in the Amhara region) to December 2017 at the University of Gondar, Boru-Meda, Debre-Markos, and Woldia Hospital were included in the study.

## Inclusion and exclusion criteria

All DR-TB patients diagnose by Xpert MTB/RIF, LPA, and solid culture-based DST were included into the study while patients diagnosed on the clinical base were excluded. To evaluate the effect of diagnostics tests on the final treatment outcome based on their sputum smear result we had excluded those patients on treatment and transfer out (**Fig 2**).

## Data collection and quality assurance

Medical charts of patients and the DR-TB unit registration book were the sources of data extraction. Data were collected using closed-ended questions by four experienced Master of Public health Students and who were trained in study procedures. After collecting the data from the registration book its consistency was re-checked from the patients' chart by clinicians working in the DR-TB treatment centers.

## Data management and analysis

After proving the completeness data entry was made by using Epi-data 4.2.00 and SPSS version 20.0 statistical software was used for cleaning, coding, recoding, and analysis. Descriptive statistics of patients' demographic and clinical characteristics were reported in the text, tables,

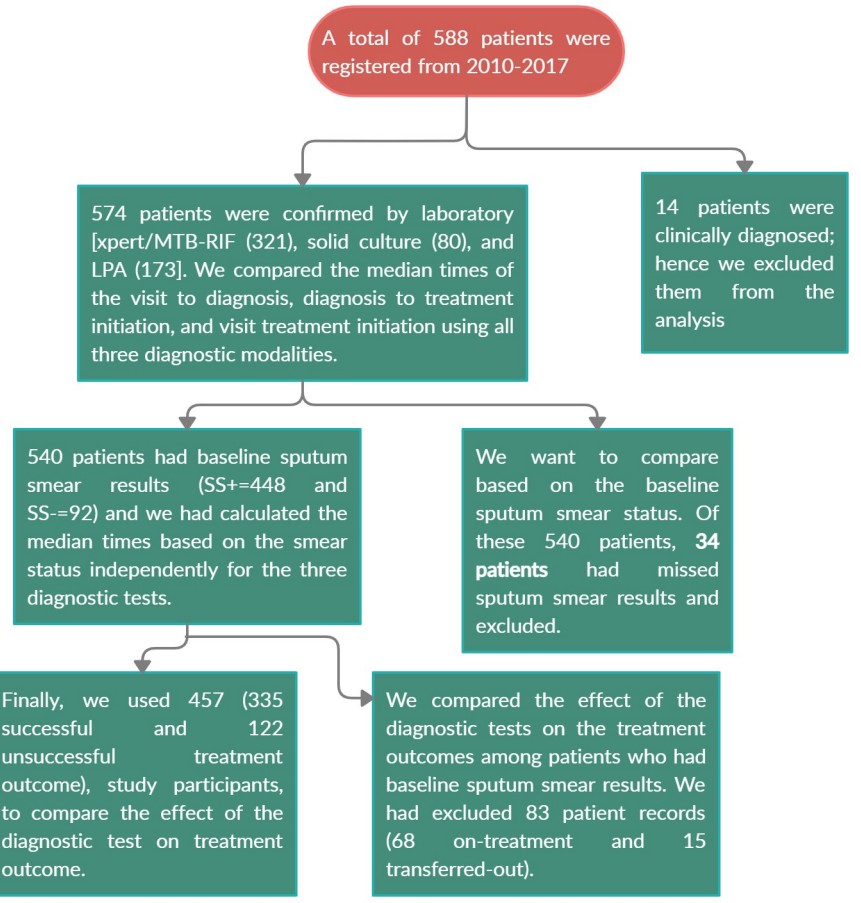

**Fig 2. A diagram depicting the inclusion/exclusion of DR-TB patients to the study.**

and graphs. The time variable and age of the patients were not normally distributed continuous variables. As a result, we summarized by their median and interquartile range (IQR). The null hypothesis (Ho) states that the distribution of time and the treatment outcomes among DR-TB are the same across categories of DR-TB genotypic diagnosis algorisms and the Alternative Hypothesis (Ha) was stated that the distribution of time and the treatment outcome of DR-TB are different across categories of DR-TB genotypic diagnosis algorisms. To test the hypothesis (the existence of time difference from: a first care-seeking visit to diagnosis, diagnosis to treatment start, and first care-seeking visit to treatment start among DR-TB diagnostic algorisms) we used the **Independent Sample Kruskal-Wallis Test**. And to determine the presence of a statistically significant association between categories (DR-TB diagnostic algorism groups and the treatment outcome), we used **Pearson's chi-square ($X^2$) or Fisher exact test**, where appropriate. The P-value was calculated and the result was considered statistically significant when it was $< 0.05$.

**Ethical Clearance.** The Ethical Review Committee of the University of Gondar approved the study. A permission and support letter were also obtained from the Amhara public health institutions, the management committee of each hospital, and TB ward heads. Thus, informed consent was waived as we received the institutional support letter. Information obtained at any course of the study was kept confidential. The data were anonymized and personal identifiers were not included and secured via password in computerized databases to ensure confidentiality. The results of this study were made accessible to clinicians for decision-making as early as available.

## Results

A total of 588 DR-TB patients were registered and start treatment from September 2010 to December 2017. Among these 14 patient records diagnosed on clinically bases were excluded. We have analyzed the different demographic, clinical, and laboratory characteristics from the remaining 574 patient records. From these, 329 (57.32%) were males. The median age was 28 (IQR = 23, 40) years. Nearly sixty percent (58.56%) of patients were from UoG hospitals. Greater than a quarter (27.87%) of them were Human Immunodeficiency Virus (HIV) co-infected. From those who had baseline sputum smear test, 448 (81.45%) had a positive Sputum Smear (SSm+) result and 92 (16.73%) has a baseline negative Sputum Smear (SSm-) result (**Table 1**).

### Median time

The overall median time and IQR in days from first care-seeking visit to diagnosis, from diagnosis to treatment, and from first care-seeking to treatment were: 2 (0,9), 3 (1,8) and 4 (1,11) for Xpert MTB/RIF; 4 (1,55), 33 (4,76) and 3 (1,12) for LPA; and 70 (18,182), 44 (9, 145) and 76 (3.75,191) for solid culture. The time was decreased in the three-time expressing dimensions in the Xpert MTB/RIF compared to the LPA and solid culture-based diagnostic algorisms. Similarly, the LPA-based diagnostic algorism has a short median time compared to that of the solid culture-based DST tool. These time differences were statistically significant across the DST methods in the independent sample Kruskal-Wallis test (**Table 2**).

The median time from diagnosis to treatment start among DR-TB patients was rapid if diagnosed with Xpert MTB/RIF followed by LPA in both SSm+ and SSm- groups. Patients with baseline SSm+ are also start treatment as early as from the SSm- (**Fig 3**).

We analyze the median time based on their baseline sputum smear result from 540 patients because 34 patients recorded were excluded due to missed smear results. To assess the significance of the time difference we used the independent sample Kruskual-Wallis test and

**Table 1. Demographic and clinical characteristics of DR-TB patients stratified by the diagnostic methods in Amhara regional state, Ethiopia.**

| | Frequency (N) and Percent (%) of DR-TB patients | | |
|---|---|---|---|
| **Variables** | **Solid Culture base N (%)** | **LPA base N (%)** | **Xpert MTB/RIF N (%)** |
| Sex | | | |
| **Male** | 50 (15.20) | 98 (29.79) | 181 (55.01) |
| **Female** | 30 (12.25) | 75 (30.61) | 140 (57.14) |
| Median age in year (IQR) | 28 (23, 40) | 28.5 (22, 38) | 28 (23,25, 40) |
| Treatment initiating and follow-up site hospital | | | |
| **University of Gondar** | 67 (19.94) | 126 (37.50) | 143 (42.56) |
| **Boru-Meda** | 7 (5.00) | 38 (27.14) | 95 (67.86) |
| **Debre-Markos** | 3 (6.52) | 9 (19.57) | 34 (73.91) |
| **Woldia** | 3 (5.77) | 0 (0.00) | 49 (94.23) |
| HIV status | | | |
| **Positive** | 18 (11.25) | 41 (25.62) | 101 (63.13) |
| **Negative** | 62 (15.16) | 130 (31.78) | 217 (53.06) |
| **Unknown** | 0 (0.00) | 2 (40.00) | 3 (60.00) |
| Baseline sputum smear result | | | |
| **Positive** | 68 (15.18) | 143 (31.92) | 237 (52.90) |
| **Negative** | 7 (7.61) | 23 (25.00) | 62 (67.39) |
| **Not recorded** | 5 (14.71) | 7 (20.59) | 22 (64.70) |
| History of TB treatment for first-line anti TB drugs | | | |
| **Yes** | 74 (15.19) | 170 (34.91) | 243 (49.90) |
| **No** | 6 (6.90) | 3 (3.45) | 78 (89.65) |

observed a statistically significant difference in all-time dimension among baseline Sputum Smear Positive (SSm+) patients and from diagnosis to treatment time in those patients with baseline Sputum Smear Negative (SSm-) results (**Table 3**).

## The impact of DR-TB diagnosis methods on treatment outcomes

A total of 457 DR-TB patients who had baseline sputum smear result records were included to assess the impacts of diagnostic tests on treatment outcome. Three hundred thirty-five (73.30%) patients had successful (cure plus completed) treatment outcomes. None of the diagnostic tests have been significantly associated with the treatment outcomes. We have assessed the relationship among diagnostic tests and treatment outcomes by using Pearson's Chi-square test ($X^2$) for those with SSm+ results and overall patient; and Fisher exact test for those with SSm- result. There was no statistically significant difference in the treatment outcomes based on the employed diagnostic methods (**Table 4**).

**Table 2. Median time from first care-seeking visit to diagnosis and treatment, and diagnosis to treatment among DR-TB patients in the Amhara region, Ethiopia.**

| | Median time (IQR) in days (N = 574) | | | Independent Sample Kruskal-Wallis Test (p-value) |
|---|---|---|---|---|
| | **Solid Culture-based (N = 80)** | **LPA-based (N = 173)** | **Xpert MTB/RIF (N = 321)** | |
| **From first care seeking visit to diagnosis** | 70(18,182) | 4(1,55) | 2(0,9) | <0.0001 |
| **From diagnosis to treatment (IQR)** | 44(9,145) | 33(4,76) | 3(1,8) | <0.0001 |
| **From first care seeking visit to treatment (IQR)** | 76 (3.75, 191) | 3(1,12) | 4(1,11) | <0.0001 |

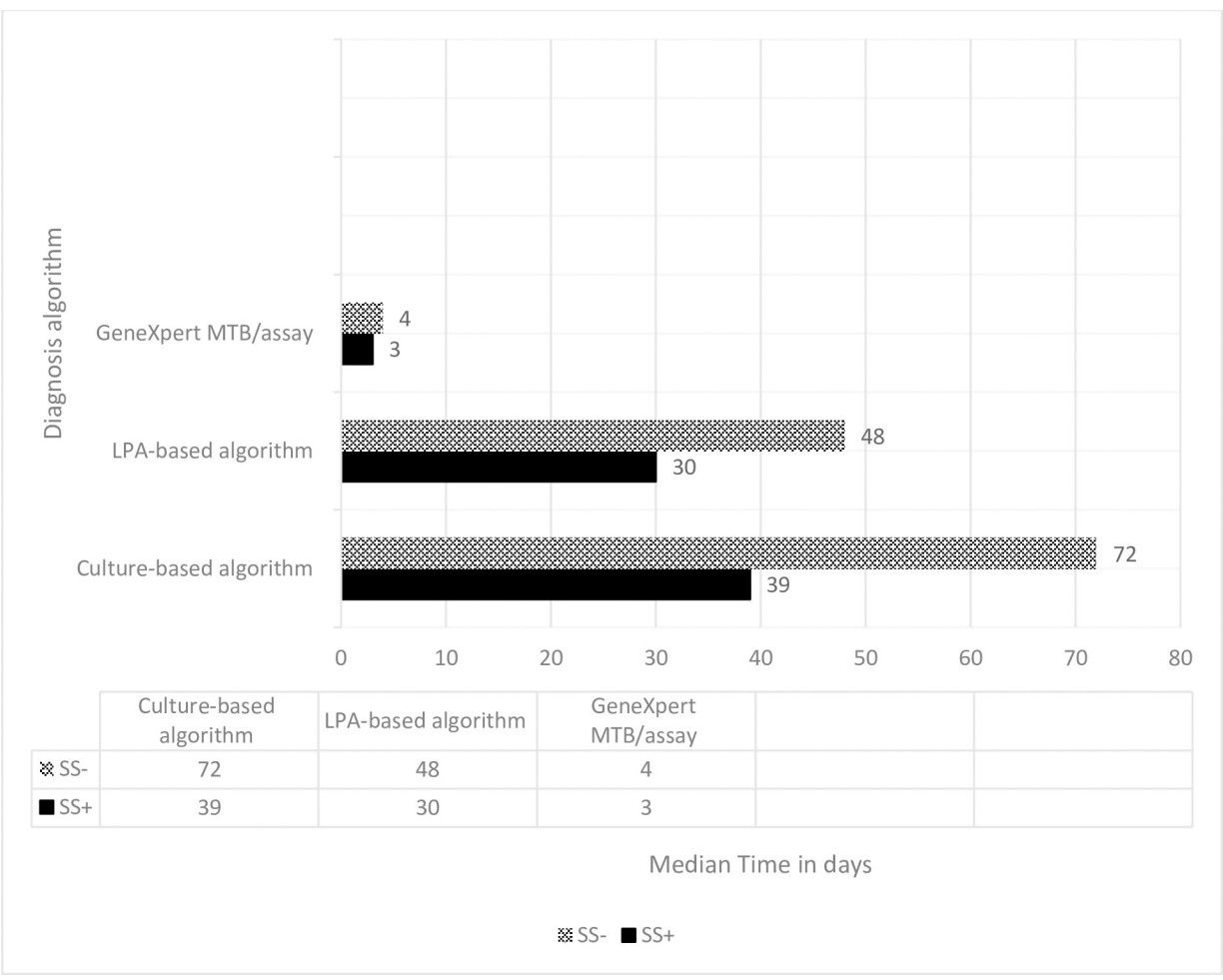

**Fig 3. Median time from diagnosis to treatment start for baseline SSm+ and SSm- DR-TB patients diagnosed with different DR-TB diagnostic algorithms.**

## Discussion

The median times in all of the three-time dimensions (from a first care-seeking visit to diagnosis, from diagnosis to treatment, and from a first care-seeking visit to treatment) was significantly early in the Xpert MTB/RIF and protracted in the solid culture-based DST. In

**Table 3. Median time from the first care-seeking visit to diagnosis and treatment, and diagnosis to treatment by their sputum smear status among DR-TB patients in Amhara region, Ethiopia (N-540).**

| | Baseline sputum smear result | Median time in days (IQR) | | | Independent sample Kruskual-Wallis (p-value) |
|---|---|---|---|---|---|
| | | Solid Culture-based DST (N = 75) | LPA-based DST (N = 166) | GeneXper MTB/RIF (N = 299) | |
| From the first care-seeking visit to diagnosis | SSm+ | 84(22,185) | 4(1,38.5) | 2(0,6.5) | <0.0001 |
| | SSm- | 14 (10, 32) | 4(1.5, 48.5) | 1.5(0.2,6.5) | 0.226 |
| From diagnosis to treatment | SSm+ | 39(8.5,145.5) | 30(4,72) | 3(1,7) | <0.0001 |
| | SSm- | 72(2,198) | 48(4,115) | 4(0,10) | 0.008 |
| From 1st visit to the treatment | SSm+ | 82(3,204) | 3(1,12.25) | 3(1,8) | <0.0001 |
| | SSm- | 41(1,8) | 3(1.5,13.5) | 6(1,46) | 0.473 |

**Table 4. Treatment outcomes of DR-TB based on the diagnostic methods stratified by their sputum smear result in the Amhara Region, Ethiopia.**

| Categories | Diagnostic method | Treatment outcome of DR-TB patients (N = 457) | | |
|---|---|---|---|---|
| | | Successful (Cured + Completed) | Unsuccessful (Failure +Lost +died) | |
| SSm+ | Xpert MTB/RIF | 135 | 43 | $X^2$ |
| | LPA | 98 | 44 | P = 0.076 |
| | Solid Culture | 55 | 11 | |
| SSm- | Xpert MTB/RIF | 28 | 15 | Fisher Exact test |
| | LPA | 14 | 8 | |
| | Solid Culture | 5 | 1 | P = 0.80 |
| Overall (SSm+ and SSm-) | Xpert MTB/RIF | 163 | 58 | $X^2$ |
| | LPA | 112 | 52 | P = 0.054 |
| | Solid Culture | 60 | 12 | |

agreement with these, multiple studies from Zimbabwe [28], South Africa [29], Tanzania [30], Nigeria [31], India [32, 33], and Russia [5] evidenced that -Xpert MTB/RIF was a rapid diagnosis tool that resulted in rapid treatment commencement. This can be explained in that the laboratory result by Xpert MTB/RIF was available with two hours which was much earlier in contrast to the LPA (48 hours) and solid Culture (in weeks) base DST methods. The other possible cause might be the need for high laboratory bio-safety and skilled personals in solid culture and LPA based DST methods whereas Xpert MTB/RIF is an easy to do diagnostic assay that can be executed with minimal training and bio-safety [34]. It could be also due to the availably of onsite diagnostic tests, from the study settings only the University of Gondar hospitals had a recently established TB solid culture test center, but the rest of the three hospitals were referring samples to other solid culture execution sites (regional public health and research laboratories). Implementation of a rapid DR-TB diagnostic test like Xpert MTB/RIF was capable of early case finding, facilitates the TB control programs, proper treatment initiations, and fruitful treatment outcomes [32].

Post diagnosis treatment delay was observed in the order of solid culture and LPA based DST tests compared to Xpert MTB/RIF in both who had baseline SSm+ and SSm- results. Similar result was reported from another study in Ethiopia [35]. And we also observed that a significant median time difference from the first care-seeking visit to the diagnosis, the diagnosis to treatment, and the first care-seeking visit to treatment in the three diagnostic tests among patients with baseline SSm+ (Table 3). This might be due to the required time to produce a result by each diagnosis method, the Xpert MTB/RIF assay will take two hours to produce results, LPA two days, and LJ culture will take weeks. It can be also due to a health facility and patient-related delays. One, there were several DR-TB patients on the waiting list to start second-line anti-TB drugs after their DR-TB diagnosis because of a shortage of beds and drugs. Two, DR-TB diagnostic tests were not available onsite to all of the DR-TB TICs in the region. Because of this, samples were collected from patients and stored in a refrigerator till transported to the sample processing sites. The first DR-TB diagnostic site in Ethiopia was the Ethiopian Public Health Institute (EPHI) located in Addis Abeba (the capital city of Ethiopia) followed by the Amhara Public Health Institute (APHI) located in Bahr-Dar, which is too far from these TICs. The sample from the patients at each TICs was collected, stored, and send to EPHI or APHI for further diagnostic tests. The days taken for sample storage, transportation, and late report of results may also produce dalliance in the diagnosis and treatment of patients. Three, the patient may delay presenting to health facilities after diagnosis. On the contrary, we didn't see a significant difference in the median time from the first care-seeking visit to the diagnosis and the first care-seeking visit to treatment among the three diagnostic tests of those

patients who had baseline SSm- results. These can be explained by the low MTB load in the SSm- patients to be detected by an Xpert MTB/RIF test. The available Gene-Xpert machine in the study area was the conventional Gene-Xpert machine which requires hundreds of bacilli load/ml of sample. But, the LJ culture-based DST can grow and detect up to 10 live bacilli/ml of sample. For those patients who give sputum or other samples for LJ culture, the result will be available within three to six weeks. Whereas, in the study area if patients had a negative Xpert MTB/RIF test result because of low bacillary load or other reasons, they will be appointed to come and re-evaluated after three weeks of taking broad-spectrum antibiotics which may delay the diagnosis and treatment of DR-TB by three or more weeks. These three or more week delays in the Xpert MTB/RIF test and the normal taking by the solid culture median may produce a comparable median time between the two test methods.

There was no statistically significant difference in terms of final treatment outcomes in all of the DR-TB diagnostic algorisms. From literatures, we found a controversial result between the different diagnostic tools and treatment outcomes. For instance findings from Tanzania [30], South Africa [23], and Brazil [36] have indicated that there was no association between the treatment outcome and diagnostic methods but a study in Russia revealed that the introduction of LPA significantly improved treatment outcome compared to LJ culture-based DST [5].

The limitation of this study was the use of historical data and due to lack of data, we didn't include the laboratory result turn round time, available health care services in each hospitals, health care worker-related data, socioeconomic status of patients, and we have also failed to analyze the effect of the drug regimens used to treat DR-TB on final treatment outcomes. The rapid laboratory result turn round time, the capacity and quality of health care facilities services, and the number of expert health care workers might favor quicker decision making, thereby shorter time to treatment initiation without being directly affected by diagnostic methods.

## Conclusion and recommendation

Xpert MTB/RIF can mitigate the transmission of DR-TB significantly via quick diagnosis and treatment initiation followed by LPA as equating to the solid culture base DST, particularly in smear-positive patients. But we didn't see a statistically significant impact in terms of treatment outcomes. Xpert MTB/RIF can be used as the first test to diagnose DR-TB by further complemented with a solid culture base DST to grasp the drug-resistance profile.

## Acknowledgments

The authors are thankful to the data collectors for the success of data collection. We were also thanks to health care workers working at the DR-TB treatment centers towards their assistance in the curation of data. We would also like to thank all the hospital administrations for their permission to review the treatment cards of DR-TB patients.

## Author Contributions

**Conceptualization:** Getahun Molla Kassa, Mehari Woldemariam Merid, Atalay Goshu Muluneh, Haileab Fekadu Wolde.

**Data curation:** Getahun Molla Kassa, Mehari Woldemariam Merid, Atalay Goshu Muluneh.

**Formal analysis:** Getahun Molla Kassa.

**Methodology:** Atalay Goshu Muluneh, Haileab Fekadu Wolde.

**Validation:** Getahun Molla Kassa, Mehari Woldemariam Merid, Haileab Fekadu Wolde.

**Writing – original draft:** Getahun Molla Kassa.

**Writing – review & editing:** Getahun Molla Kassa, Mehari Woldemariam Merid, Atalay Goshu Muluneh, Haileab Fekadu Wolde.

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
