## [Decision Letter · Decision Letter 0]

10 Aug 2020

PONE-D-20-18404

Comparing the impact of Genotypic Based Diagnostic Algorithm on Time to Treatment Initiation and Treatment Outcomes among Drug-Resistant Tuberculosis Patients in Amhara Region, Ethiopia

PLOS ONE

Dear Dr. Merid,

Thank you for submitting your manuscript to PLOS ONE. After careful consideration, we feel that it has merit but does not fully meet PLOS ONE’s publication criteria as it currently stands. Therefore, we invite you to submit a revised version of the manuscript that addresses the points raised during the review process.

Please submit your revised manuscrip. If you will need more time than this to complete your revisions, please reply to this message or contact the journal office at plosone@plos.org. Please include the following items when submitting your revised manuscript:

We look forward to receiving your revised manuscript.

Kind regards,

Frederick Quinn

Academic Editor

PLOS ONE

Journal Requirements:

2. Please provide the date(s) when you accessed the data used for the study.

3. In your ethics statement in the Methods section and in the online submission form, please provide additional information about the data used in your retrospective study. Specifically, please ensure that you have discussed whether all data were fully anonymized before you accessed them and/or whether the IRB or ethics committee waived the requirement for informed consent. If patients provided informed written consent to have data from their medical records used in research, please include this information.

4. We suggest you thoroughly copyedit your manuscript for language usage, spelling, and grammar. If you do not know anyone who can help you do this, you may wish to consider employing a professional scientific editing service.  

Reviewers' comments:

Reviewer's Responses to Questions

**Comments to the Author**

1. Is the manuscript technically sound, and do the data support the conclusions?

Reviewer #1: Partly

Reviewer #2: Yes

2. Has the statistical analysis been performed appropriately and rigorously? 

Reviewer #1: Yes

Reviewer #2: Yes

3. Have the authors made all data underlying the findings in their manuscript fully available?

Reviewer #1: Yes

Reviewer #2: No

4. Is the manuscript presented in an intelligible fashion and written in standard English?

Reviewer #1: Yes

Reviewer #2: Yes

5. Review Comments to the Author

Reviewer #1: Dear editor,

Review: ‘Comparing the impact of Genotypic Based Diagnostic Algorithm on Time to Treatment Initiation and Treatment Outcomes among Drug-Resistant Tuberculosis Patients in Amhara Region, Ethiopia’.

PONE-D-20-18404

The topic under discussion is very relevant and important to understand the effect of the molecular diagnostics in the early diagnosis and treatment initiation, and its impact on patient-level treatment outcome. However, in the present form, the manuscript needs major revision and correction in its methodology and analysis. Given the authors satisfactorily address the comments I would recommend it for publication.

Reviewer #2: This manuscript present important information on comparison of three diagnostic algorithm on time of treatment initiation of DR-TB and its treatment outcome in Amhara region of Ethiopia. This is information important for TB control sector and this manuscript can be considered for the publication, provided that authors revised as per comments given.

Comments

- It is good if authors include map of study area with indicated location of study hospitals on the map.

- Table titles are very long and needs to revise and make short and informative

- In discussion part: line 232-234. "There was no statistically significant difference in terms of final treatment outcomes in all of the DR-TB diagnostic algorisms. From literatures, we found a controversial result between the different diagnostic tools and treatment outcomes"

The reason for this controversy might be confounding factor in your analysis and did author checked for confounding factors in there statistical analysis?

6. PLOS authors have the option to publish the peer review history of their article (what does this mean?). If published, this will include your full peer review and any attached files.

Reviewer #1: No

Reviewer #2: No

---

## [Author Response · Author response to Decision Letter 0]

19 Oct 2020

Date August/25/2020

Rebuttal letter

PONE-D-20-18404

Comparing the impact of Genotypic Based Diagnostic Algorithm on Time to Treatment Initiation and Treatment Outcomes among Drug-Resistant Tuberculosis Patients in Amhara Region, Ethiopia

Mehari Woldemariam Merid

To PLOS ONE

Dear all,

We are very pleased to thank you for providing very relevant, constructive, and critical comments and questions about the manuscript. Based on your comments and questions, we authors have tried to review the document in detail. The point by point responses are presented in two columns as reviewers’ comments and authors’ response as follows. The detail revision and changes we made in the main document is described as track change in the document attached separately. We hope the revised manuscript will better suit the journal, but we are happy to consider farther revisions.

Reviewers comment Authors response

Major comments: 

Abstract 

The methods and the result section in the abstract need to be revised based on the detailed comments in the methods section. The major comments are around median time to treatment and categorization of the three comparison groups and treatment outcomes. Answer: Dear Reviewer, thank you very much for your fruitful comment.

According to your comment, we have corrected the questions you have raised and incorporated on the main documents. We have also made clarifications or re-write on the topics which need re-writings or clarifications.

Line 41, ’in smear-positive patients’, your discussion shows the opposite, and your data does not fully support this. Please revise. Answer: Thank you, dear reviewer.

Dears, this conclusion was made based on our result as indicated in Table 3, which shows Xpert MTB/RIF significantly reduced the median time from first care-seeking visit to diagnosis, diagnosis to treatment initiation, and first care-seeking visit to treatment initiation among smear-positive DR-TB as compared to the other diagnostic methods. The problem here might be the way we have discussed, now we have tried to discuss it clearly in the revised manuscript. Page: 15 &16, Line: 222-241

Line 30, 32, 35, 39 and 42, GeneXpert is the instrument (or the platform) and Xpert MTB/RIF is the assay (test), please correct this throughout the manuscript. Answer: Thank you, reviewer, for your comment.

Agreeing to your comment we have corrected it throughout the entire document.

Introduction 

Line 58-63: your paper is on drug-resistant TB, the references you used from 6 to 9 are on drug-sensitive TB and thus do not apply to your study. Please replace them with the relevant references. Answer: Thank you, dear reviewer, for the comment.

Accordingly, we have agreed with your comment. Currently, we have corrected and replaced with references done on DR-TB in the main document. Page: 4, Line: 57, 59 & 60

Line 72-72, sentence starting ‘As compared to culture….’. The sentence is not correct, the overall sensitivity of Xpert never been that high. If you meant, among smear-positive that might be possible. If so, qualify the sentence accordingly. Answer: Thank you, dear reviewer, for the comment.

Dear reviewer, we have re-read the paper and they have done on presumptive TB patients not among smear-positive patients. They have also compared the Xpert MTB/RIF test with the MGIT culture test and reported as the Xpert MTB/RIF had 99.1% sensitivity and 96% specificity to detect DR-TB cases compared to MGIT culture. We have also read and tried to find the reasons for this finding in their discussion part, but they didn’t put any further justifications.

Line 80 and 82, reference number 6 is on drug-sensitive TB and thus does not apply to your study. Please replace it as appropriate Answer: Thank you, dear reviewer.

Dear reviewer, now we have corrected it according to your recommendations on the main document. 

Page: 5, Line: 75 & 77

Methods

In the methods part, the following are missing: 

1. Since the three tests are usually conducted one after another, it is not clear how the three distinctive comparison groups were categorized. For instance, if a sample tested positive by Xpert MTB/RIF with Rifampicin Resistance, then LPA needs to be conducted to confirm for Isoniazid resistance. Please clarify, how the three exclusive comparison groups (Xpert MTB/RIF, LPA, and Culture) were created, whether the country was using three different algorithms at different times? Answer: Thank you, dear reviewer, for raising this concern.

Sure! We have to share your concerns. Xpert MTB/RIF was introduced in the Amhara region, Ethiopia by September 2013, before that culture and LPA was used alternatively to diagnose DR-TB. Even after the introduction of Xpert MTB/RIF in 2013, culture and LPA were also used to diagnose DR-TB because of the limited Xpert MTB/RIF test access. Currently, as you have stated these three tests are done one after the other. Once the patient becomes resistant to Rifampicin by Xpert MTB/RIF then first and second line LPA was performed to see further resistance for INH, FLQ, and injectable. The culture was done to monitor the treatment response and for further phenotypic DST performance.

2. The definition of terms for treatment outcomes is not appearing in the methods part. Also, the categorization of treatment outcome needs to be reconsidered. A comparison of outcome is helpful when categorized as successful (cured and treatment completion) and unsuccessful (failure, died, and lost to follow-up). You may use references Johnston et al PLOS ONE 2009 or Evan W Orenstein BS et al, the Lancet infectious disease, 2009). Since we don’t know the final treatment outcome, transferred out needs to be excluded when you compare treatment outcome. Please revise Table 4. Answer: Dear Reviewer, thank you very much for your suggestion.

Accordingly, we have included the definition of treatment outcome and re-categorized as successful (cured and treatment completion) and unsuccessful (failure, died, and lost to follow-up) and cited the recommended reference as per your suggestion in the method part of the main document. Based on this definition of treatment outcome we have also re-analyzed and revised Table 4 on the main manuscript.

Page: 7, Line: 115-119

3. The measurement of the median time, particularly the ‘time to treatment initiation’ is not clear. It is not immediately clear how do you define a diagnosis, and thus time to diagnosis? Please clarify and include in your methods. Time from the first visit to diagnosis and time from diagnosis to treatment are subsets of time from the first visit to treatment. Based on this, the median time from initial visit to treatment does not seem to right or at least the numbers do not make sense in Table 3. Please clarify how time to treatment is less than that of time to diagnosis and time to treatment (from diagnosis). Answer: Thank you, dear reviewer, for inquiring a clarity on these critical issues.

Accordingly, we clarified that point in the main document as follows. Time from a first care-seeking visit to diagnosis was defined as the total date taken from the patient visit the health facility for care to the time of their tuberculosis diagnosis. Whereas time from diagnosis to treatment initiation was calculated from the date taken from DR-TB diagnosis to anti-TB treatment initiation; and time from first care-seeking visit to treatment initiation was labeled as the date from the patient comes to the health facility for seeking care to the date the patient start anti-TB treatment. Page: 7, Line: 110-115.

Dears, we have seen and recognized that the median time from the first care-seeking visit to treatment was lower than the median time from diagnosis to treatment in the LPA method in Table 3 when we analyze and write the result initially. Again, we have critically examined the data one by one based on your comment and we assure you that these findings were true. The reason why this happened was, the data from diagnosis to treatment (IQR) was negatively skewed while the data from the first care-seeking visit to diagnosis and a first care-seeking visit to treatment was positively skewed. As you know from the basic statistics, the median time was taken at the midpoint of all observation. If the data is negatively skewed the median will be higher but if it is a positively skewed data the median will be lower. Hence, the data nature stated here will explain the difference.

4. MDR-TB treatment regimens are one of the major factors affecting treatment outcomes, and there is no mention of MDR-TB treatment regimens in the study period (2010-2017). The WHO changed the recommendation to drug-resistant treatment in 2011 and 2016. Also, if there was a change in treatment regimen between 2010 and 2017 it needs to be clarified. Answer: Thank you, Dear reviewer.

Dears, Of course, the used MDR-TB treatment regimen has a great role on the treatment outcomes of patients. Similarly to the WHO, Ethiopia changed the recommended MDR-TB treatment in 2014, 2018, and 2019. During the study period, the treatment regimen of MDR-TB were changed at once in 2014. Even, in the same guideline, there were two or more recommended regimens for MDR-TB treatment based on the patients' clinical and laboratory parameters. Dear reviewer, there are a bunch of publishing reports on the predictors of final treatment outcomes of MDR-TB in the study area including the used MDR-TB treatment regimens. Plus to this, our objectives were aimed to determine the median time on the diagnosis, treatment initiation, and identify their effect on treatment outcome of different DR-TB diagnosis test. So, the question you have will be appropriate if our study was aimed to identify the factors of MDR-TB treatment outcome. Because of the above mentioned two reasons (it is out of our objectives and the presence of published reports), we differed to see the effect of MDR-TB treatment regimens in our study.

5. Line 100-101, the sentence starting ‘over 90% of DR_TB….’ needs a reference. Answer: Thank you, Dear reviewer.

Now we have cited a reference in the main document. Page: 6, Line: 96

6. Line 123, it is not immediately clear why the smear result was an excursion criterion. The diagnosis of drug-resistant TB is not based on the smear. Please clarify if you still want to keep it as exclusion criterion. Answer: Thank you, dear reviewer, for your clarity concern.

The exclusion criteria we have used in this study were those patients diagnose based on clinical criteria. We use sputum smear result as the exclusion criteria while we were comparing the different median time among patients who has sputum smear result and when we have to try to see the effect of the diagnosis test on the final treatment outcome stratified by their sputum smear results. To make it more clear now we have re-write in the main document as follows. “All DR-TB patients diagnosed by Xpert MTB/RIF, LPA, and culture-based DST were included into the study while patients diagnosed on the clinical basis were excluded. To evaluate the effect of diagnosis tests on the final treatment outcome stratified by their sputum smear result, we had excluded those patients who were on treatment and transferred out.” Moreover, now it is presented in Figure 2 to make it more clearly in this regard.

7. You need to include another table indicating the distribution of patients among the four hospitals, and which diagnostic tool used in which hospital. Answer: Thank you, reviewer, for your comment.

We have already included the number of patients and the used diagnostic methods in each hospital in Table 1 as the variable name “Treatment initiating and follow-up site hospital”. Table 1, Page: 10

8. You need to include which culture method was used, solid culture versus liquid culture. Answer: Thank you, dear reviewer, for your critical comment.

The solid (Löwenstien-Jensen or LJ) culture media were the used culture method in this study and we have corrected it in the main documents.

Results 

1. Table 1, the median age in years…the 16.75, 28.5 and 17 are not IQR, this needs correction Answer: Thank you, reviewer, for your critical comment.

Dears, we apologized for the fault we made, we erroneously inserted the SD instead of the IQR. Now, we have corrected and incorporated it into the main document. Table 2, Page: 11

2. Table 1, ‘smear result not recorded’ should be taken out based on your exclusion criteria Answer: Thank you, reviewer, for the comment.

Dear reviewer, Table 1 was presented to show the descriptive of patients’ sociodemographic and clinical characteristics. We didn’t use sputum smear result for exclusion at the outset. Because of this, we have presented the sputum smear status of all the study patients. We excluded those patients who didn’t have smear results when we analyze the data to see the presence of median time and outcome difference among smear-positive and negative patients. To make clearer on the exclusion and inclusion criteria now we have included a diagram in the main document. Please refer to Figure 2.

3. Tables 2 and 3 need revision based on the comments on time to treatment in the methods. Answer: Thank you, dear reviewer, for the comment.

Dears, we have addressed this question in question number three in the method part. Figure 2.

4. Line 152, since the 34 don’t have recorded results, based on your exclusion criteria 14 should be 48 (14 plus 34). Answer: Thank you again, reviewer.

Dears, to make clear on the inclusion and exclusion criteria, we have re-write and clarified the main documents and Figure 2. 

5. Line 181, you did well by excluding the 34. That is the same reason you should exclude the 34 throughout. Answer: Dear reviewer, thank you very much.

We intended to compare the impact of diagnostic tests on the final treatment outcomes of DR-TB patients based on their baseline sputum smear status. To do so, 34 patient sputum smear result were missed, 68 patients were on treatment, and six patients had no recorded sputum smear result as well as they were also on treatment. Because of the above-listed reasons we excluded from the analysis. 

But at the revised manuscript based on the recommendations, you made before we have corrected as follows in the main document. “A total of 457 DR-TB patients who had baseline sputum smear result records were included to assess the impacts of diagnostic tests on treatment outcomes.” 

To make clear how the number of patients comes to 457 we have added a diagram in the revised manuscript as Figure 2.

6. Line 194, ‘six patients were on both’…. this is not clear when you state both treatments, which is which? Answer: Thank you, dear reviewer!

Dear reviewer, We have addressed this question in question number 5 above.

7. Table 4, need revision based on the recommended treatment outcome categorization (successful versus unsuccessful treatment outcome). See the previous comment in the methods section. Answer: Thank you, reviewer, for your comment.

Dears, we have revised it according to your comments. Table 4, Page: 14

Discussions 

Line 229-231, the sentence starting with ‘the rationale for this….’ This does not sound true. In fact, the opposite is true. The lower bacteria load delays time to culture positivity and the DST that follows culture. Please clarify or correct. Answer: Thank you, reviewer, for your comment.

Dear reviewer, we have read your opinion and justification. We have tried to clear more in this regard in the main documents as follows. “This might be due to the required time to produce a result by each diagnosis method, the Xpert MTB/RIF assay will take two hours to produce results, LPA two days, and LJ culture will take weeks. On the contrary, we didn’t see a significant difference in the median time from the first care-seeking visit to the diagnosis and the first care-seeking visit to treatment among the three diagnostic tests of those patients who had baseline SS- results. These can be explained by the low MTB load in the SS- patients to be detected by an Xpert MTB/RIF test. The available Gene-Xpert machine in the study area was the conventional Gene-Xpert machine which requires hundreds of bacilli load/ml of sample. But, the LJ culture-based DST can grow and detect up to 10 live bacilli/ml of sample. Those patients who give sputum or other samples for LJ culture, the result will be available within three to six weeks. Whereas, in the study area if patients had a negative Xpert MTB/RIF test result because of low bacillary load or other reasons, they will be appointed to come and re-evaluated after three weeks of taking broad-spectrum antibiotics which may delay the diagnosis and treatment of DR-TB by three or more weeks. These three or more week delays in the Xpert MTB/RIF test and the normal taking by the solid culture median may produce a comparable median time between the two test methods.” Page: 15 & 16, Line: 222-241.

Line 232-237, this discussion is likely to change based on the comments in the methods and result on treatment outcomes (Table 4). Please revise it accordingly. Answer: Thank you, reviewer, for your comment.

Dears, based on your suggestion we have re-analyze to see the effect of diagnostic tests on treatment outcome after recoding the treatment outcome in successful and unsuccessful and we found insignificant results, which is similar to the previous reports. Because of this, we kept the discussions unchanged.

Limitations: line 240, why was ‘the health care facilities’ included as missing data. Didn’t you know the four hospitals you studied? In fact, would have been best if you included the result on the comparison between the hospitals in median times and treatment outcomes. Also, the distribution of patients among the four hospitals is missing. Wiser if you include those. As a limitation, you also need to include the lack of data on socioeconomic status and mode of treatment whether DOTS, at facilities or community based, as some of the factors potentially affecting treatment outcomes. Answer: Thank you, dear reviewer, once again.

Sorry for the lack of clarification. We have also added your recommendations as a limitation in the revised manuscript. We wrote the health facilities as missing data just to mean the detailed characteristics of each hospital service/setups like the availability and quality of medical care, nutritional supports, drug supplies, and so on. Dear Reviewer, the mode of treatment was health facilities based DOTs throughout the whole treatment period. We have already tried to show the distribution of patients among the four hospitals in Table 1 of the main document. Regarding the effect of other factors on the final treatment outcome, there are several published reports on other factors potentially affecting the treatment outcome in the study area and it is also out of our objectives. Page: 16, Line: 248-253.

Conclusions and recommendation 

Line 247…’particularly in smear-positive patients’, your discussion shows the opposite, and your data does not fully support this. Please revise. Answer: Thank you, reviewer, for raising this important concern.

Dears, we have reviewed the result of our study and tried to improve the discussion part as seen from the main document. Disagreeing with you, the conclusion we made was based on our result you can see Table 3, which shows the Xpert MTB/RIF test significantly reduced the median time among smear-positive DR-TB as compared to the LPA and LJ culture-based diagnostic tests. The problem in the previous manuscript might be, we didn’t thoroughly discuss why this was happening. In the recent manuscript, we have revised the discussion related to the median time among smear-positive and negative DR-TB patients and tried to put the possible rational. Page: 15 and 16 , Line: 222-241.

Line 247-248 may change after your reanalysis of treatment outcomes (Table 4). Please revise it accordingly. Answer: Thank you, dear reviewer, for your productive comments.

Dear reviewer, as per your notes we have re-analyzed the outcome and the finding was similar to the earlier report, because of this we kept it as it is.

Minor comments: 

Abstract 

Line 27 and 28, delete ‘with p-value’ and ‘finally, the p-value was reported’. Mentioning these may not be that necessary. Answer: Thank you, reviewer, for your comment.

Dears, we have removed it from the main manuscript.

Introduction 

Line 54 and 55, tuberculosis is already abbreviated, thus use it as TB moving forward. There are lots of places in the manuscript like this, please correct accordingly. Answer: We thank you, dear reviewer, for your valuable suggestions.

As a result of your suggestions, we have corrected it all over the main document.

Line 57, LPA need to be written in full since first-time use. Answer: Thank you, dear reviewer, for your comment.

Convening to your comment we have written LPA in full as Line Probe Assay (LPA) in the manuscript.

Line 58, replace the sentence with…. ‘Shorten time to diagnosis, and reduced delay to treatment among DR-TB, and improved final treatment outcomes.’ Answer: Dear reviewer, we thank you for your c, valuable suggestion.

According to your suggestion, we have substituted the sentence in the main document.

Line 62, GeneXpert is the instrument (or the platform) and Xpert MTB/RIF is the assay (test), please correct this throughout the manuscript. Answer: We gratefully thank you, dear reviewer, for your comment.

Dears, We agree with your valuable comments and corrected it now throughout the main manuscript.

Line 90, replace ‘we are intended’ with ‘we aim’ Answer: Thank you very much, dear reviewer, for your comment.

We take your comment and corrected it in the manuscript.

Methods 

Line 105, add a comma after historically, Answer: Thank you, reviewer, for your comment.

According to your suggestion, we have added the comma in the main document.

Line 116, replace the sentence in parenthesis with ‘data difference from the diagnosis to initiation of treatment.’ Answer: Thank you, reviewer, for your comment.

Dears, we have corrected the main document as follows. “The date difference from the DR-TB diagnosis to initiation of anti-TB treatment”.

Result 

Line 154, replace ‘three-hundred twenty-nine’ with 329 Answer: Thank you, reviewer, for your comment.

We have replaced now

Line 155, remove ‘with’ after 28 and add old after ‘years’ Answer: Thank you, reviewer, for the comment.

We have done accordingly in the main document.

Line 160, replace ‘drug-resistant tuberculosis’ with DR-TB. Answer: Dear reviewer thank you for your comment.

We have corrected it in the main manuscript.

Line 161, replace ‘DR-TB diagnosis’ with ‘diagnostic’ Answer: Thank you, reviewer, for your comment.

We have replaced it now in the main document.

Line 164-167, the use of semi-colon and comma does not flow well. Replace the sentences with ‘The overall median time and IQR in days from first care-seeking visit to diagnosis, from diagnosis to treatment, and from first care-seeking to treatment were: 2 (0,9), 3 (1,8) and 4 (1,11) for Xpert MTB/RIF; 4 (1,55), 33 (4,76) and 3 (1,12) for LPA; and 70 (18,182), 44 (9, 145) and 76 (3.75,191) for culture.’ …your numbers of course will be corrected per the revised analysis and result. Answer: Thank you very much, dear reviewer, for your valuable suggestions.

According to your proposed, we have re-write it in the main documents.

Line 167, remove respectively. Answer: Thank you, reviewer, for the comment.

Now, we have removed it from the main document

Line 194, use a text for a number in a single digit. Replace 6 with ‘six’ Answer: Dear reviewer thank you for your comment.

We have replaced it

Line 194, add ‘s’ after the patient (patients) and replace was with were. Six patients were….. Answer: Thank you, reviewer, for your comment.

We have corrected it accordingly in the main document.

Discussion 

Line 212, replace ‘diagnosis TD-TB tool’ with ‘diagnostic tool’ Answer: Thank you, reviewer, for your comment.

We have replaced it with the documents.

Line 216, replace GeneXpert with ‘Xpert MTB/RIF’ Answer: Thank you, dear reviewer, for your comment.

We have replaced it throughout the documents.

Line 216, Xpert MTB/RIF test is not simple. It is a very complex process, but easy to perform. Thus replace ‘a simple diagnostic assay’ with ‘an easy to do diagnostic assay’. 

Line 239, add ‘data’ after missing. Answer: Dear reviewer, thank you very much for your comment.

We have corrected it in the main document as per the given comments.

Acknowledgment 

Line 253, remove ‘s’ from datas… Answer: Thank you, reviewer, for your comment.

We have removed it in the main document.

Authors’ contribution 

Replace ‘Dta’ with ‘Data’. Answer: Thank you, reviewer, for your comment.

We have replaced Dta with data in the main document.

References 

Line 343, Ref # 24, seems not well written. Please revise. Answer: thank you, Dear reviewer.

We have corrected it and incorporated it on the documents.

Line 363-4, the reference seems missing the publication number. Please revise. Answer: Thank you, Dear Reviewer.

Now, we have added the publication number on the main documents. 

It is good if authors include map of study area with indicated location of study hospitals on the map. Answer: Thank you, dear reviewer, once again.

Accordingly, we have included a map of the study area. Figure 1. 

Table titles are very long and needs to revise and make short and informative Answer: Thank you, Dear Reviewer.

Accordingly, we have tried to shorten the titles of tables. Table 1, 2, 3, & 4, page: 9, 11, 12 & 13.

In discussion part: line 232-234. "There was no statistically significant difference in terms of final treatment outcomes in all of the DR-TB diagnostic algorisms. From literatures, we found a controversial result between the different diagnostic tools and treatment outcomes"

The reason for this controversy might be confounding factor in your analysis and did author checked for confounding factors in there statistical analysis? Answer: Dear reviewer, thank you.

As you have noted, these controversial findings were noticed from published papers, not from our findings. Some reports say there is a significant difference and some reports also said there is no. These might be because of different reasons like the confounding variables, methodological, and measurement issues.

---

## [Decision Letter · Decision Letter 1]

1 Dec 2020

PONE-D-20-18404R1

Comparing the impact of Genotypic Based Diagnostic Algorithm on Time to Treatment Initiation and Treatment Outcomes among Drug-Resistant Tuberculosis Patients in Amhara Region, Ethiopia

PLOS ONE

Dear Dr. Merid,

Thank you for submitting your manuscript to PLOS ONE. After careful consideration, we feel that it has merit but does not fully meet PLOS ONE’s publication criteria as it currently stands. Therefore, we invite you to submit a revised version of the manuscript that addresses the points raised during the review process.

Please submit your revised manuscript. If you will need more time than this to complete your revisions, please reply to this message or contact the journal office at plosone@plos.org. Please include the following items when submitting your revised manuscript:

We look forward to receiving your revised manuscript.

Kind regards,

Frederick Quinn

Academic Editor

PLOS ONE

Reviewers' comments:

Reviewer's Responses to Questions

**Comments to the Author**

1. If the authors have adequately addressed your comments raised in a previous round of review and you feel that this manuscript is now acceptable for publication, you may indicate that here to bypass the “Comments to the Author” section, enter your conflict of interest statement in the “Confidential to Editor” section, and submit your "Accept" recommendation.

Reviewer #1: (No Response)

2. Is the manuscript technically sound, and do the data support the conclusions?

Reviewer #1: Partly

3. Has the statistical analysis been performed appropriately and rigorously? 

Reviewer #1: Yes

4. Have the authors made all data underlying the findings in their manuscript fully available?

Reviewer #1: Yes

5. Is the manuscript presented in an intelligible fashion and written in standard English?

Reviewer #1: Yes

6. Review Comments to the Author

Reviewer #1: Review: ‘Comparing the impact of Genotypic Based Diagnostic Algorithm on Time to Treatment Initiation and Treatment Outcomes among Drug-Resistant Tuberculosis Patients in Amhara Region, Ethiopia’.

PONE-D-20-18404_R1

The topic under discussion remains very relevant and important to understand the effect of the molecular diagnostics in the early diagnosis and treatment initiation, and its impact on patient-level treatment outcome. In the revised form, the manuscript still needs major revision and correction in its methodology, analysis and discussion.

Major comments:

Abstract

The methods and the discussion section in the abstract need to be revised based on the detailed comments in the main manuscript. The major comments are around methods are clarifying how the three comparison categorizes were created.

Introduction

None

Methods

Comment ‘since the three tests are usually conducted one after another, it is not clear how the three distinctive comparison groups were categorized. For instance, if a sample tested positive by Xpert MTB/RIF with Rifampicin Resistance, then LPA needs to be conducted to confirm for Isoniazid resistance’

The below response for the above comment needs to be summarized and included in the methods section. Without it the readers would still get confused in the process and the conclusion the authors attempted to reach.

Xpert MTB/RIF was introduced in the Amhara region, Ethiopia by September 2013, before that culture and LPA was used alternatively to diagnose DR-TB. Even after the introduction of Xpert MTB/RIF in 2013, culture and LPA were also used to diagnose DR-TB because of the limited Xpert MTB/RIF test access. Currently, as you have stated these three tests are done one after the other. Once the patient becomes resistant to Rifampicin by Xpert MTB/RIF then first and second line LPA was performed to see further resistance for INH, FLQ, and injectable. The culture was done to monitor the treatment response and for further phenotypic DST performance.

Results

1. Table 2, the median time from to diagnosis (33) and to treatment (3) discrepancies still need further discussion. The earlier is more the health system factor and the latter is more patient related factor such as delayed presentation for treatment after diagnosis. See comments on the discussion part. age in years…the 16.75, 28.5 and 17 are not IQR, this needs correction

2. Table 3, since the table is by smear status, either correct the heading to 540 or if you want to keep 574 include the median times for unknown smears too. Also, if you add 68,143 and 237 gives you 448, not 540 or 574. Please correct or clarify.

The result under LPA ss+ and ss-, 30 versus 48; and culture 84 versus 14 raised concern. E.g. one would not expect ss+ to have higher median time that ss- for obvious reasons. The higher the bacillary load the faster for the culture to convert to positive result, thus time to diagnosis should be lesser.

Discussions

Line 231-232, ‘This was 232 supported by a study from Ethiopia [35]’. How does ref #35 support you result? Rather, indicate the result how similar or different it is.

Line 133-134, ‘And we also observed that a significant delay in terms 233 of time from the first care-seeking visit to the diagnosis and the first care-seeking visit to 234 treatment among patients with baseline SSm+ result’. This is only true for culture, and not indicated in your results. In addition, the comparison in your result (Table 3), is between diagnostics (culture, LPA and Xpert MTB/RIF), not ss+ versus ss-. To make your discussion appropriate you need the later analysis.

Line 134-136, again, the median time to diagnosis from first visit is more of a health system related factor and median time to treatment is more of a patient related factor such as delay to present to health facilities after diagnosis. Thus, you need to clarify and discuss the points in the detail and explanation they deserve.

Line 136-139, same comment as above applies, clarify your discussion points and make it more plausible. As it is lacks sufficient details.

Line 242-249, This discussion with ss-, you are contradicting yourself where your earlier points state ss+ has delays than ss-. Please revise and clarify your discussion here.

In addition, some discussion points are lacking as to why diagnosis to treatment in LPA is longer than first visit to treatment. LPA and Xpert has only one or two days difference in terms of turnaround time versus culture that may take months. Your clarification on skewedness does not fully explain.

Limitation, please include not presenting the DR-TB regimens as a limitation since one cannot discuss treatment outcome without describing the treatment regimen. It is even better, if you include in the methods section the clarification you provided in your response in this regard.

Conclusions and recommendation

Once you edited the discussion based on the comments provided you may revise the conclusion too.

Minor comments:

Methods

Figure 2, please edit the table to professional formatting. E.g. your arrows are not straight, not aligned and their sizes are not equal. Also, in the second box – replace ‘expert’ to Xpert MTB/RIF.

Result

Line 163, usually we use numbers when 10 and above. Replace fourteen with 14.

7. PLOS authors have the option to publish the peer review history of their article (what does this mean?). If published, this will include your full peer review and any attached files.

Reviewer #1: No

---

## [Author Response · Author response to Decision Letter 1]

14 Jan 2021

Date: January 14, 2021

Rebuttal letter

PONE-D-20-18404R1

Comparing the impact of Genotypic Based Diagnostic Algorithm on Time to Treatment Initiation and Treatment Outcomes among Drug-Resistant Tuberculosis Patients in Amhara Region, Ethiopia

Mehari Woldemariam Merid

To PLOS ONE

Dear all,

We the authors of this manuscript are pleased to thank the journal editors and the reviewers for revising the manuscript and giving your valuable and constructive comments and suggestions that help to improve the manuscript. We have made a rigorous revision of the manuscript as per your questions and comments. We have included the point by point response in the table below framed as reviewers’ comment/question and authors’ response. The detailed revision and changes we made in the main document are prepared with track changes in the document attached separately. We expect that the revision we made will enable the manuscript to fit the journal. We are happy to receive additional revision if any that would have merit in improving the manuscript.

Reviewers comments Authors Response 

Major Comments 

Abstract 

The methods and the discussion section in the abstract need to be revised based on the detailed comments in the main manuscript. The major comments are around methods are clarifying how the three comparison categorizes were created. Dear reviwer, as per your sugestions we have revised the method parts of the abstract. Since there was no discussion part in the abstract we didn’t need to revise it. 

Methods 

Comment ‘since the three tests are usually conducted one after another, it is not clear how the three distinctive comparison groups were categorized. For instance, if a sample tested positive by Xpert MTB/RIF with Rifampicin Resistance, then LPA needs to be conducted to confirm for Isoniazid resistance’

The below response for the above comment needs to be summarized and included in the methods section. Without it the readers would still get confused in the process and the conclusion the authors attempted to reach.

Xpert MTB/RIF was introduced in the Amhara region, Ethiopia by September 2013, before that culture and LPA was used alternatively to diagnose DR-TB. Even after the introduction of Xpert MTB/RIF in 2013, culture and LPA were also used to diagnose DR-TB because of the limited Xpert MTB/RIF test access. Currently, as you have stated these three tests are done one after the other. Once the patient becomes resistant to Rifampicin by Xpert MTB/RIF then first and second line LPA was performed to see further resistance for INH, FLQ, and injectable. The culture was done to monitor the treatment response and for further phenotypic DST performance. Thanks, Dear Reviewer

We had summarized and incorporated it into the method parts of the manuscript as follows. “Culture-based DST and LPA was used alternatively to diagnose DR-TB in Ethiopia until 2013. Latter in 2013, Xpert MTB/RIF was introduced in the study area to diagnose RR-TB. Currently, only TB culture-based DST and LPA sites are available in the Amhara region, which is located in Bahr-Dar and Gondar. The access of these three diagnosis tests varies from site to site, because of this and some other reasons even after the introduction of Xpert MTB/RIF, culture and LPA were also used as the first test to diagnose DR-TB. Currently, those three tests are done serially. Once the patient was diagnosed to have RR-TB by Xpert MTB/RIF then first and the second line LPA was performed to see further resistance for Isoniazid, Floroqunolols, and injectable ant-TB drugs, and culture was done to monitor the treatment response and when further phenotypic DST is necessary.”

Results 

1. Table 2, the median time from to diagnosis (33) and to treatment (3) discrepancies still need further discussion. The earlier is more the health system factor and the latter is more patient related factor such as delayed presentation for treatment after diagnosis. See comments on the discussion part. age in years…the 16.75, 28.5 and 17 are not IQR, this needs correction Sure, there were some patients presented with advanced and debilitating TB disease and highly indexed for DR-TB. In that case, the clinical panel team at the DR-TB center may start the standard DR-TB treatment clinically for those patients when the laboratory result is pending and revised their decision when the laboratory result was available. 

We didn’t get the point of what you want to raise related to age. There was no age in table two. Age was described in table one as median and IQR.

2. Table 3, since the table is by smear status, either correct the heading to 540 or if you want to keep 574 include the median times for unknown smears too. Also, if you add 68,143 and 237 gives you 448, not 540 or 574. Please correct or clarify. Dear Reviewer, you are right the sum of 68,143 and 237 gives 448, not 540 or 574. This is a typo error. The error was, we wrongly writing only the number of SS+ (448) patients diagnostic in each tests, now we have corrected it by adding 92 SSm- DR-TB patients.

3. The result under LPA ss+ and ss-, 30 versus 48; and culture 84 versus 14 raised concern. E.g. one would not expect ss+ to have higher median time that ss- for obvious reasons. The higher the bacillary load the faster for the culture to convert to positive result, thus time to diagnosis should be lesser. The result under LPA doesn’t raise concern because the time in SS+ 30 days was shorter than SS- 48 days. 

As you have stated there might be concerns on culture-based DSTs, for ss+ 84 days verses for ss- 14 days. This might be due to, at the first two years year of the DR-TB diagnosis and treatment history, the diagnosis was made from high-risk patients like patients with smear-positive by referring the sample to the Ethiopian public health institute located in Addis Abeba. At this time there may delays due to transportation, late reporting results.

Discussions 

Line 231-232, ‘This was 232 supported by a study from Ethiopia [35]’. How does ref #35 support you result? Rather, indicate the result how similar or different it is. Dear Reviewer, we have corrected it in the main manuscript based on your suggestions as follows. “Similar result was reported from another study in Ethiopia (35).”

Line 133-134, ‘And we also observed that a significant delay in terms 233 of time from the first care-seeking visit to the diagnosis and the first care-seeking visit to 234 treatment among patients with baseline SSm+ result’. This is only true for culture, and not indicated in your results. In addition, the comparison in your result (Table 3), is between diagnostics (culture, LPA and Xpert MTB/RIF), not ss+ versus ss-. To make your discussion appropriate you need the later analysis. Dear reviewer, the problem we have observed here is our poor sentence construction, which doesn’t communicate the message about what we want to forward. We have tried to clarify the sentence as follows in the main document. “And we also observed that a significant median time deference from the first care-seeking visit to the diagnosis, the diagnosis to treatment, and the first care-seeking visit to treatment in the three DR-TB diagnostic tests among patients with baseline SSm+ (Table 3).”

Line 134-136, again, the median time to diagnosis from first visit is more of a health system related factor and median time to treatment is more of a patient related factor such as delay to present to health facilities after diagnosis. Thus, you need to clarify and discuss the points in the detail and explanation they deserve. Dear reviewer, we accept your suggestions and had discussed it as follows. “This might be due to the required time to produce a result by each diagnosis method, the Xpert MTB/RIF assay will take two hours to produce results, LPA two days, and LJ culture will take weeks. It can be also due to a health facility and patient-related delays. One, there were several DR-TB patients on the waiting list to start second-line anti-TB drugs after their DR-TB diagnosis because of a shortage of beds and drugs. Two, DR-TB diagnostic tests were not available onsite to all of the DR-TB TICs in the region. Because of this, samples were collected from patients and stored in a refrigerator till transported to the sample processing sites. The first DR-TB diagnostic site in Ethiopia was the Ethiopian Public Health Institute (EPHI) located in Addis Abeba (the capital city of Ethiopia) followed by the Amhara Public Health Institute (APHI) located in Bahr-Dar, which is too far from these TICs. The sample from the patients at each TICs was collected, stored, and send to EPHI or APHI for further diagnostic tests. The days taken for sample storage, transportation, and late report of results may also produce dalliance in the diagnosis and treatment of patients. Three, the patient may delay presenting to health facilities after diagnosis.”

Line 136-139, same comment as above applies, clarify your discussion points and make it more plausible. As it is lacks sufficient details. 

Line 242-249, This discussion with ss-, you are contradicting yourself where your earlier points state ss+ has delays than ss-. Please revise and clarify your discussion here. Dear reviewer, we agree with you, this contradicting discussion was established because of the poor sentence construction in the SSm+. Now we have re-write the sentence to make the discussion clear in the main manuscript.

In addition, some discussion points are lacking as to why diagnosis to treatment in LPA is longer than first visit to treatment. LPA and Xpert has only one or two days difference in terms of turnaround time versus culture that may take months. Your clarification on skewedness does not fully explain. Sure, these were because of the accessibility of tests and there was also a platform for rapid reporting of results for Xpert MTB/RIF tests.

Limitation, please include not presenting the DR-TB regimens as a limitation since one cannot discuss treatment outcome without describing the treatment regimen. It is even better, if you include in the methods section the clarification you provided in your response in this regard. Thank you, dear reviewer, we had included in the limitation that this study lacks to assess the effect of DR-TB regimens on final treatment outcomes in the main manuscript. 

Conclusions and recommendation 

Once you edited the discussion based on the comments provided you may revise the conclusion too. Dear reviewer, we dint find reasons that can enable us to revise our conclusion.

Minor comments: 

Methods 

Figure 2, please edit the table to professional formatting. E.g. your arrows are not straight, not aligned and their sizes are not equal. Also, in the second box – replace ‘expert’ to Xpert MTB/RIF. Thank you, dear reviewer, we have edited figure two according to your comments.

Result 

Line 163, usually we use numbers when 10 and above. Replace fourteen with 14. Thank you dears, we have replaced it.

---

## [Decision Letter · Decision Letter 2]

29 Jan 2021

Comparing the impact of Genotypic Based Diagnostic Algorithm on Time to Treatment Initiation and Treatment Outcomes among Drug-Resistant Tuberculosis Patients in Amhara Region, Ethiopia

PONE-D-20-18404R2

Dear Dr. Merid,

We’re pleased to inform you that your manuscript has been judged scientifically suitable for publication and will be formally accepted for publication once it meets all outstanding technical requirements.

Kind regards,

Frederick Quinn

Academic Editor

PLOS ONE

Additional Editor Comments (optional):

Reviewers' comments:

Reviewer's Responses to Questions

**Comments to the Author**

1. If the authors have adequately addressed your comments raised in a previous round of review and you feel that this manuscript is now acceptable for publication, you may indicate that here to bypass the “Comments to the Author” section, enter your conflict of interest statement in the “Confidential to Editor” section, and submit your "Accept" recommendation.

Reviewer #1: All comments have been addressed

2. Is the manuscript technically sound, and do the data support the conclusions?

Reviewer #1: Yes

3. Has the statistical analysis been performed appropriately and rigorously? 

Reviewer #1: Yes

4. Have the authors made all data underlying the findings in their manuscript fully available?

Reviewer #1: Yes

5. Is the manuscript presented in an intelligible fashion and written in standard English?

Reviewer #1: Yes

6. Review Comments to the Author

Reviewer #1: All the comments have been addressed satisfactorily, and the English has improved. The few remining minor comments are: 1) to make sure that the clarifications you provided to the comments are included in the discussion. E.g. response for the 30 versus 48 and 84 versus 14, and so on. 2) In the limitation part, replace 'we have failed' with ' due to incomplete data we were not able to analyze'.

7. PLOS authors have the option to publish the peer review history of their article (what does this mean?). If published, this will include your full peer review and any attached files.

Reviewer #1: No

---

## [Editor Report · Acceptance letter]

5 Feb 2021

PONE-D-20-18404R2 

Comparing the impact of Genotypic Based Diagnostic Algorithm on Time to Treatment Initiation and Treatment Outcomes among Drug-Resistant Tuberculosis Patients in Amhara Region, Ethiopia 

Dear Dr. Merid:

I'm pleased to inform you that your manuscript has been deemed suitable for publication in PLOS ONE. Congratulations! Your manuscript is now with our production department. 

Kind regards, 

on behalf of

Dr. Frederick Quinn 

Academic Editor

PLOS ONE